# The relationship between alpha power and heart rate variability commonly seen in various mental states

**Tomoya Kawashima** [1]*, **Honoka Shiratori**[2], **Kaoru Amano**[2]

1 Department of Psychological Science, College of Informatics and Human Communication, Kanazawa Institute of Technology, Nonoichi, Ishikawa, Japan, 2 Graduate School of Information Science and Technology, The University of Tokyo, Tokyo, Japan

* kawashima-t@neptune.kanazawa-it.ac.jp

**Data Availability Statement:** The data underlying the results presented in the study are available from OSF (https://osf.io/aw4z7/).

**Funding:** SPS Grant-in-Aid for Early-Career Scientists (JP20K14274 and JP 23K17171). The

## Abstract

The extensive exploration of the correlation between electroencephalogram (EEG) and heart rate variability (HRV) has yielded inconsistent outcomes, largely attributable to variations in the tasks employed in the studies. The direct relationship between EEG and HRV is further complicated by alpha power, which is susceptible to influences such as mental fatigue and sleepiness. This research endeavors to examine the brain-heart interplay typically observed during periods of music listening and rest. In an effort to mitigate the indirect effects of mental states on alpha power, subjective fatigue and sleepiness were measured during rest, while emotional valence and arousal were evaluated during music listening. Partial correlation analyses unveiled positive associations between occipital alpha2 power (10–12 Hz) and nHF, an indicator of parasympathetic activity, under both music and rest conditions. These findings underscore brain-heart interactions that persist even after the effects of other variables have been accounted for.

## Introduction

Homeostatic regulation of the body's internal state and its impact on brain function are recognized to shape behavior and cognition [1, 2]. Consequently, a functional assessment of the brain-heart interplay can provide valuable insights into human physiological and behavioral dynamics. Alpha oscillations are prominently measured in EEG [3] and will originate from thalamo-cortical interactions [4]. While closely associated with perception and cognition [5–11], they are also known to reflect changes in alertness. Specifically, mental states characterized by relaxation or drowsiness are associated with increases in alpha power. For example, an increase in alpha power can be observed in a meditative state, indicating a state of relaxed alertness [12, 13]. According to Shaw [14], an increase in alpha power is associated with inward-directed attention, while a decrease in alpha power is associated with outward-directed attention.

Heart Rate Variability (HRV) is the time variation between successive heartbeats, which is a beat-to-beat variation of R-R intervals. HRV analysis can assess the state of the autonomic

funders had no role in study design, data collection and analysis, decision to publish, or preparation of the manuscript.

**Competing interests:** The authors have declared that no competing interests exist.

nervous system [15]. The frequency feature in HRV is thought to be related to the dynamics of the sympathetic and parasympathetic nervous systems. While the low frequency (LF) band (0.04–0.15 Hz) reflects both sympathetic and vagal responses, the high frequency (HF) band (0.15–0.4 Hz) may be associated with parasympathetic control of the heart [15]. The relationship between HRV and emotional arousal has been studied. For example, Iwanaga et al. [16] showed that excited music increased the LF component of HRV, suggesting a decrease in parasympathetic nervous system activation. Another line of research reported that relaxing music increases parasympathetic nervous system activity, as reflected in HRV [17]. Thus, the quantification of HRV parameters can serve as an index of the autonomic nervous system's status.

The direct correlations between human EEG alpha oscillations and heart rate variability (HRV) have been the subject of numerous studies using a variety of tasks, with inconsistent results. For example, decreased alpha power is associated with increased sympathetic activity due to decreased attention [18], mental fatigue [19], and, paradoxically, decreased parasympathetic activity due to increased emotional arousal [20]. Consequently, alpha power can be influenced by a variety of factors, making it difficult to establish a direct relationship between alpha oscillations and heart rate activity.

This study seeks to investigate the brain-heart interplay that is consistently observed across different mental states. Given that emotional stimuli affect both EEG and HRV [21–23], this study investigates the relationship between EEG and heart rate during the experience of emotional music. Furthermore, based on the relationship between EEG and HRV during the resting state in the same subjects, the brain-heart interplay commonly observed across different mental states (i.e., music listening and resting) is investigated. Specifically, we measure subjective arousal and emotional valence during music listening, as well as fatigue and sleepiness during rest. By analyzing these effects in the brain-heart interaction, this study determines common brain-heart interactions across different mental states. In addition, alpha oscillations are bifurcated into alpha1 and alpha2, which have been differentially associated with physiological states [24] and mental functions [25]. This approach allowed us to study the common brain-heart interplay across different biological states.

## Materials and methods

### Participants

Seventeen healthy volunteers participated in the study; all were male. One participant was removed because there was excessive noise in his EEG data, and another was removed because he did not properly understand the instruction (i.e., he inappropriately rated his mental state on the visual analog scales (VAS)). Finally, 15 participants, aged 20 to 23 years, were included in the analyses. The study was approved by the ethics and safety committee of the University of Tokyo (approval number: 21–34). A written informed consent was obtained from all participants, and all study methods were conducted according to the relevant guidelines and regulations. The experiment was carried out from the 25th of November to the 10th of December 2021.

### Stimuli

The experimental stimuli were selected to cover all quadrants of the emotional valence-arousal space. Twelve 3-minute music pieces (stimuli) that could evoke emotional responses were selected. For the eight music files, six 30s music clips were taken from the MER database [26] and combined into a 3-minute music piece. Because the MER database included a substantial amount of music that was unfamiliar to the participants, such as ethnic music, we

supplemented it with four additional music stimuli that were familiar to them and likely to evoke emotion, based on Naji et al. [27].

## Experimental procedure

Before and after the music sessions, EEG (electroencephalogram) and ECG (electrocardiogram) were recorded during a 3-minute eye-closed rest. After each 3-minute session, the participants were required to rate their fatigue and sleepiness using VAS based on the methods of Ishii et al. [19] and Lee et al. [28].

In the music condition, the overall experiment was divided into 12 blocks. Each block began with a 3-minute music presentation during which EEG and ECG were recorded while participants closed their eyes. After listening to the music, the participants completed their arousal and valence ratings using the self-assessment manikin (SAM).

## Psychophysiological measures

**EEG.** EEG signals were digitally recorded using a StarStim (Neuroelectrics, Barcelona, Spain). Eight-channel dry electrodes were placed in a Neuroelectrics cap in positions F3, F4, C3, C4, P3, P4, O1, and O2 of the 10–20 International Electrode Placement System. EEG was sampled at 500 Hz and analyzed using MNE-Python (v0.24.1: [29]).

An offline bandpass filter from 2 to 40 Hz was applied. The data were re-referenced to all electrodes. The continuous EEG data were then segmented into 2-second segments with no overlap. Epochs that exceeded a voltage threshold of 150 μV (absolute) were excluded from further analyses. A fast Fourier transform was then used to calculate the EEG power spectra with a frequency resolution of 0.5 Hz using the Welch method. The frequency band was divided into alpha1 (8–10 Hz) and alpha2 (10–12 Hz). The P3/4 and O1/2 electrodes were used for analysis because the 8–12 Hz signals are prominent in the parieto-occipital region [30].

**ECG.** ECG signals were digitally recorded from a StarStim (Neuroelectrics) at 500 Hz with an ECG cable extension attached to the subject's right wrist and left ankle. ECG analyses were conducted using pyHRV (v0.4.0: [31]). After the automatic detection of the R peaks, the inter-beat interval between R peaks was calculated. Then, a spectral analysis was performed using a Welch method to quantify the HRV in the LF (0.04–0.15 Hz) and HF (0.15–0.4 Hz). We calculated normalized LF and HF power, defined as the relative power of each component in proportion to total power minus VLF (very low frequency) [32]: normalized LF (nLF) as the LF power in normalized units LF/(total power − VLF) × 100 and normalized HF (nHF) as the HF power in normalized units HF/(total power − VLF) × 100. The nLF and nHF mainly reflect sympathetic and parasympathetic responses, respectively. Normalization can eliminate much of the substantial variability in raw HRV spectral power, both within and across subjects [33].

**Graphical modeling.** The target variables served as the nodes of the undirected graph were created, and the partial correlations corresponding to each edge were then calculated. The edge with the lowest absolute value of the partial correlation was repeatedly eliminated, stopping when the Akaike information criterion increased.

## Results

### Resting state

**Subjective change in fatigue and sleepiness.** VAS scores of subjective fatigue after the task ($M$ = 49.4, $SD$ = 18.6, $CV$ (coefficient of variation) = 0.38) were similar to those before the task ($M$ = 38.1, $SD$ = 19.4, $CV$ = 0.51) ($t$ (14) = 2.00, $p$ = .066, $d$ = 0.51). No difference was

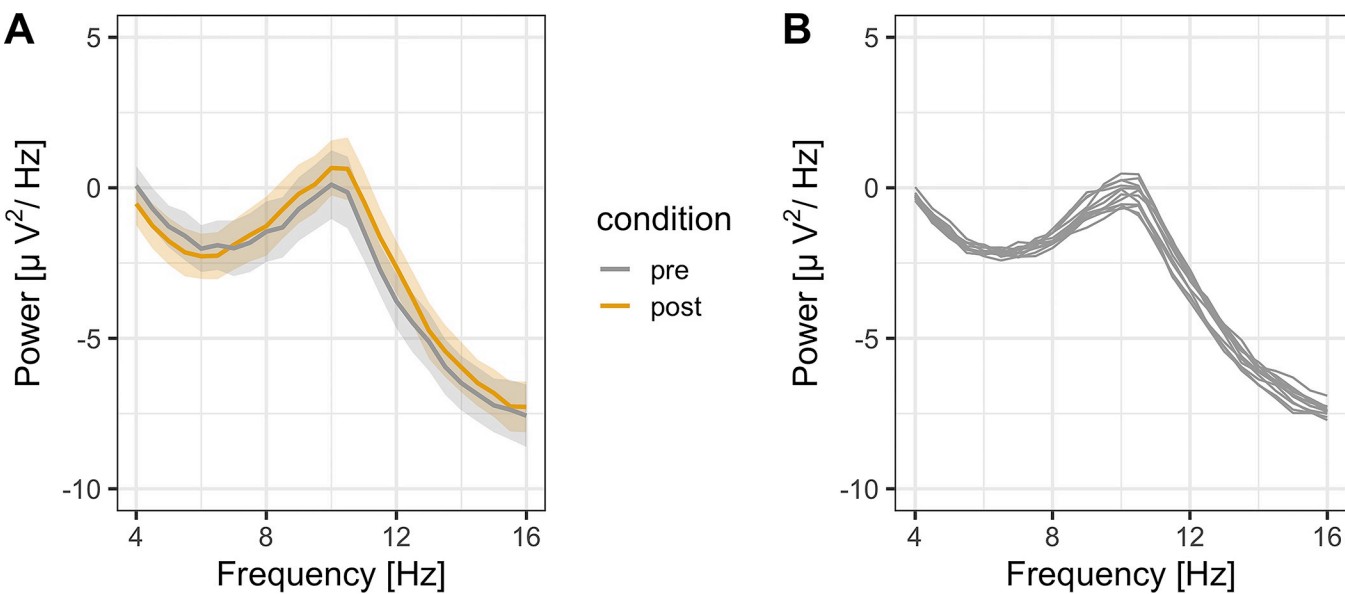

**Fig 1.** A. Power spectrum density plot of the eye-closed resting state EEG averaged over participants (electrodes: P3/4, O1/2), showing the peak alpha band (8–12 Hz) (the shaded area represents the standard error of the mean). B. Power spectrum density plot during the music exposure with eyes closed averaged over participants (electrodes: P3/4, O1/2) for 12 music stimuli. Each line represents each music stimulus, showing the peak alpha band for each stimulus.

observed in VAS scores of sleepiness before ($M$ = 49.3, $SD$ = 26.4, $CV$ = 0.54) and after the task ($M$ = 50.6, $SD$ = 20.3, $CV$ = 0.40) ($t$ (14) = 0.22, $p$ = .826, $d$ = 0.06).

**EEG.** Fig 1 shows the power spectrum density of the EEG for resting (Fig 2A), showing clear alpha peaks. The alpha1 (8–10 Hz) and alpha2 (10–12 Hz) power for pre- and post-task conditions, respectively, were calculated. Paired $t$-test showed no significant difference in the alpha1 and alpha2 power between pre- and post-tasks (alpha1, Fig 2A: $t$ (14) = 1.02, $p$ = .327, $d$ = 0.26; alpha2, Fig 2B: $t$ (14) = 1.70, $p$ = .111, $d$ = 0.44).

**HRV.** We observed significant difference in nLF between before ($M$ = 39.1, $SD$ = 13.5, $CV$ = 0.35) and after the task ($M$ = 60.9, $SD$ = 15.4, $CV$ = 0.25) ($t$ (14) = 3.03, $p$ = .009, $d$ = 1.57). Similarly, nHF showed significant difference before ($M$ = 40.2, $SD$ = 15.4, $CV$ = 0.38) and after the task ($M$ = 59.8, $SD$ = 15.4, 0.26) ($t$ (14) = 2.38, $p$ = .0.32, $d$ = 1.23).

**Music exposure.** *Subjective arousal and valence for music stimuli.* A one-way ANOVA shows no arousal or valence rating differences among songs (arousal: $F$ (11, 154) = 1.05, $p$ = .404, $\eta_p^2$ = 0.07; valence: $F$ (11, 154) = 1.05, $p$ = .404, $\eta_p^2$ = 0.07). These results suggest that the musical stimuli used in the present study may not induce subjective changes common across participants (Table 1).

*EEG.* Fig 1B shows the power spectrum density of the EEG for music exposure, showing clear alpha peaks for each music type. During the music exposure, alpha1 was not modulated by the music (alpha1: $F$ (11, 154) = 1.74, $p$ = .110, $\eta_p^2$ = .11). Alpha 2 was modulated by the music ($F$ (11, 154) = 2.39, $p$ = .009, $\eta_p^2$ = .15). Although the main effect was significant, subsequent multiple comparison showed no significant difference among pairs of means.

*HRV.* We discovered that different music styles affected HRV ($F$ (11, 154) = 2.31, $p$ = .012, $\eta_p^2$ = .14). Although the main effect was significant, subsequent multiple comparison showed no significant difference among pairs of means (Table 1).

**Correlation analysis.** *Resting state.* As noted above, no significant differences were found in subjective reports or alpha power when compared at rest before and after the task,

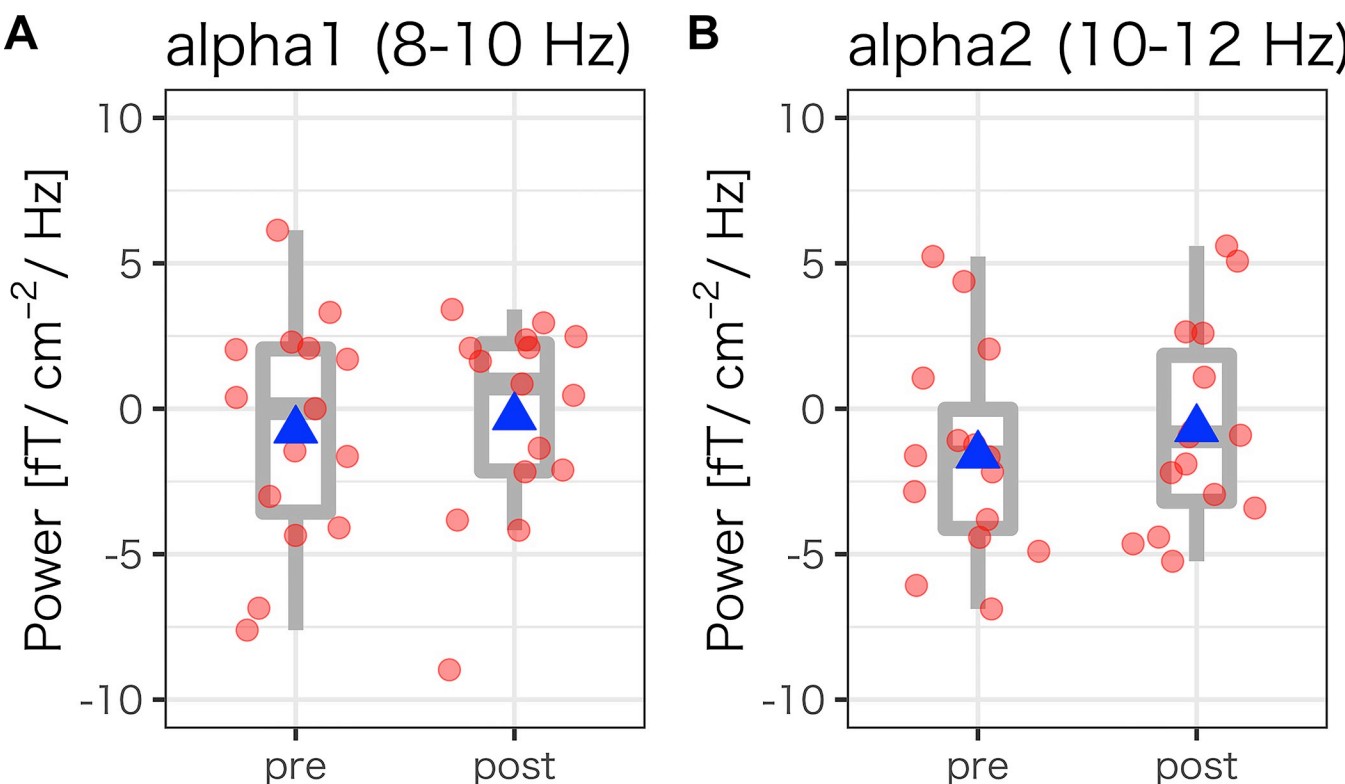

**Fig 2.** EEG power of (A) alpha1 and (B) alpha2 scores during resting state for pre- and post-task conditions, respectively. Each dot represents each participant. Blue triangles show the mean EEG power.

suggesting that changes in these variables common across participants did not exist. Here, we calculated the coefficient of variation (CV) to assess the relationship between the index and their variability relative to the mean (Table 1). The coefficient of variation is the standard deviation divided by the mean. As can be seen from some CVs in these indices that are larger than 1.0, there appears to be individual variation in the pre- and post-rest changes in these indices. Here, the differences between the pre- and post-task measurements were calculated and

**Table 1. Average, standard deviation and the coefficient of variation of subjective ratings (arousal, valence), nHF, alpha 1 and alpha 2 for each song stimuli.**

|  | arousal | | | valence | | | nHF | | | alpha 1 | | | alpha 2 | | |
|---|---|---|---|---|---|---|---|---|---|---|---|---|---|---|---|
|  | Ave. | SD | CV | Ave. | SD | CV | Ave. | SD | CV | Ave. | SD | CV | Ave. | SD | CV |
| song 1 | 4.1 | 2.0 | 0.5 | 4.1 | 2.0 | 0.5 | 59.8 | 19.0 | 0.3 | -0.5 | 4.1 | 7.7 | -0.9 | 3.5 | 3.9 |
| song 2 | 4.1 | 1.5 | 0.4 | 4.1 | 1.5 | 0.4 | 55.5 | 11.1 | 0.2 | -0.5 | 3.6 | 7.8 | -1.2 | 3.2 | 2.6 |
| song 3 | 4.7 | 1.2 | 0.3 | 4.7 | 1.2 | 0.3 | 53.8 | 16.1 | 0.3 | -0.5 | 3.7 | 7.7 | -1.2 | 3.2 | 2.6 |
| song 4 | 4.5 | 1.8 | 0.4 | 4.5 | 1.8 | 0.4 | 58.6 | 17.6 | 0.3 | -0.5 | 3.7 | 7.9 | -0.9 | 3.4 | 3.7 |
| song 5 | 3.9 | 1.5 | 0.4 | 3.9 | 1.5 | 0.4 | 50.7 | 13.9 | 0.3 | -0.8 | 3.6 | 4.6 | -1.2 | 3.1 | 2.6 |
| song 6 | 4.7 | 1.4 | 0.3 | 4.7 | 1.4 | 0.3 | 54.9 | 15.8 | 0.3 | -1.0 | 3.8 | 3.8 | -1.2 | 3.3 | 2.8 |
| song 7 | 3.7 | 1.2 | 0.3 | 3.7 | 1.2 | 0.3 | 57.4 | 11.7 | 0.2 | -1.1 | 3.7 | 3.3 | -1.2 | 3.4 | 2.8 |
| song 8 | 3.6 | 1.1 | 0.3 | 3.6 | 1.1 | 0.3 | 52.5 | 16.2 | 0.3 | -0.9 | 4.1 | 4.7 | -1.6 | 3.5 | 2.2 |
| song 9 | 4.9 | 1.3 | 0.3 | 4.9 | 1.3 | 0.3 | 49.9 | 17.2 | 0.3 | -1.1 | 3.8 | 3.5 | -1.7 | 3.4 | 2.1 |
| song 10 | 3.9 | 1.7 | 0.5 | 3.9 | 1.7 | 0.5 | 45.4 | 15.5 | 0.3 | -1.1 | 3.6 | 3.3 | -2.0 | 3.1 | 1.6 |
| song 11 | 4.3 | 1.3 | 0.3 | 4.3 | 1.3 | 0.3 | 48.9 | 17.4 | 0.4 | -1.1 | 3.7 | 3.2 | -1.9 | 3.4 | 1.8 |
| song 12 | 4.1 | 1.4 | 0.3 | 4.1 | 1.4 | 0.3 | 57.1 | 10.8 | 0.2 | -1.3 | 3.3 | 2.6 | -2.0 | 2.9 | 1.5 |

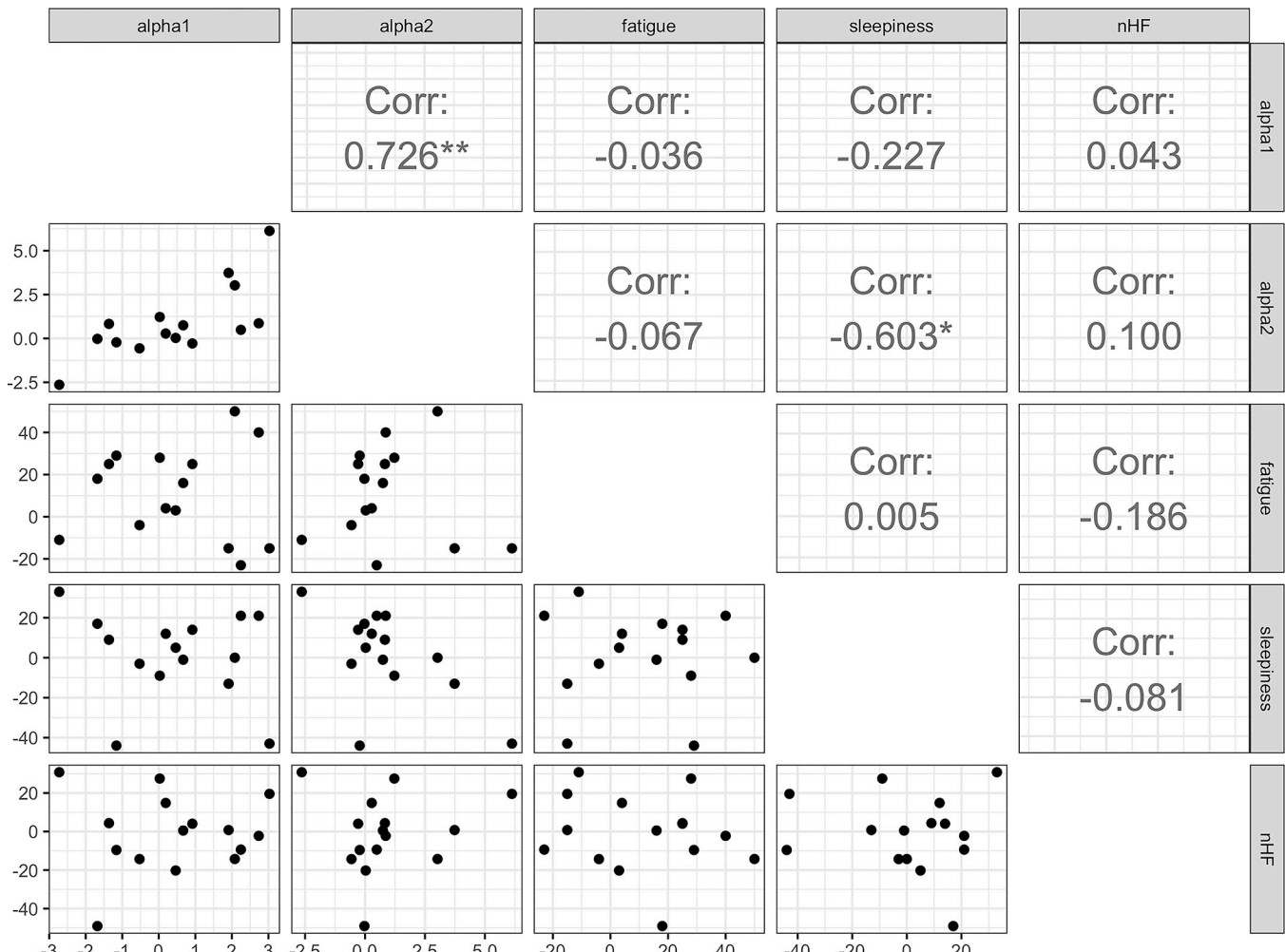

**Fig 3. Pearson correlations between subjective measures and EEG power of alpha1 and alpha2 during resting state.** Each dot represents each participants' value, which were calculated by subtraction of pre- and post-task measurements.

correlation analysis was performed between them. The results are shown in Fig 3. A positive correlation was found between alpha 1 and alpha 2 power ($r$ = .73 (95% CI[.34, .90]), $t$ (13) = 3.81, $p$ = .002). There was also a negative correlation between alpha 1 power and sleepiness ($r$ = -.60 (95% CI[-.13, -.85]), $t$ (13) = 2.73, $p$ = .017). No significant correlations were found between any of the other measurements.

*Music exposure*. As noted above, no significant differences in subjective reports and HRV were found when comparing the means of each measurement across music types, suggesting that, as with the resting-state measurements, changes in these variables common across participants did not exist. Here, a correlation analysis was performed by calculating the means of the biometric indices and subjective reports across music types. The results are shown in Fig 4. A significant positive correlation was found between alpha 1 and alpha 2 power ($r$ = .75 (95% CI [.38, .91]), $t$ (13) = 4.03, $p$ = .001). No significant correlations were found between the other indicators.

**Graphical modeling.** The correlation analysis above examines a linear relationship between two variables. However, it does not consider the indirect effects of other variables. In other words, the correlations observed above may be spurious correlations due to confounding

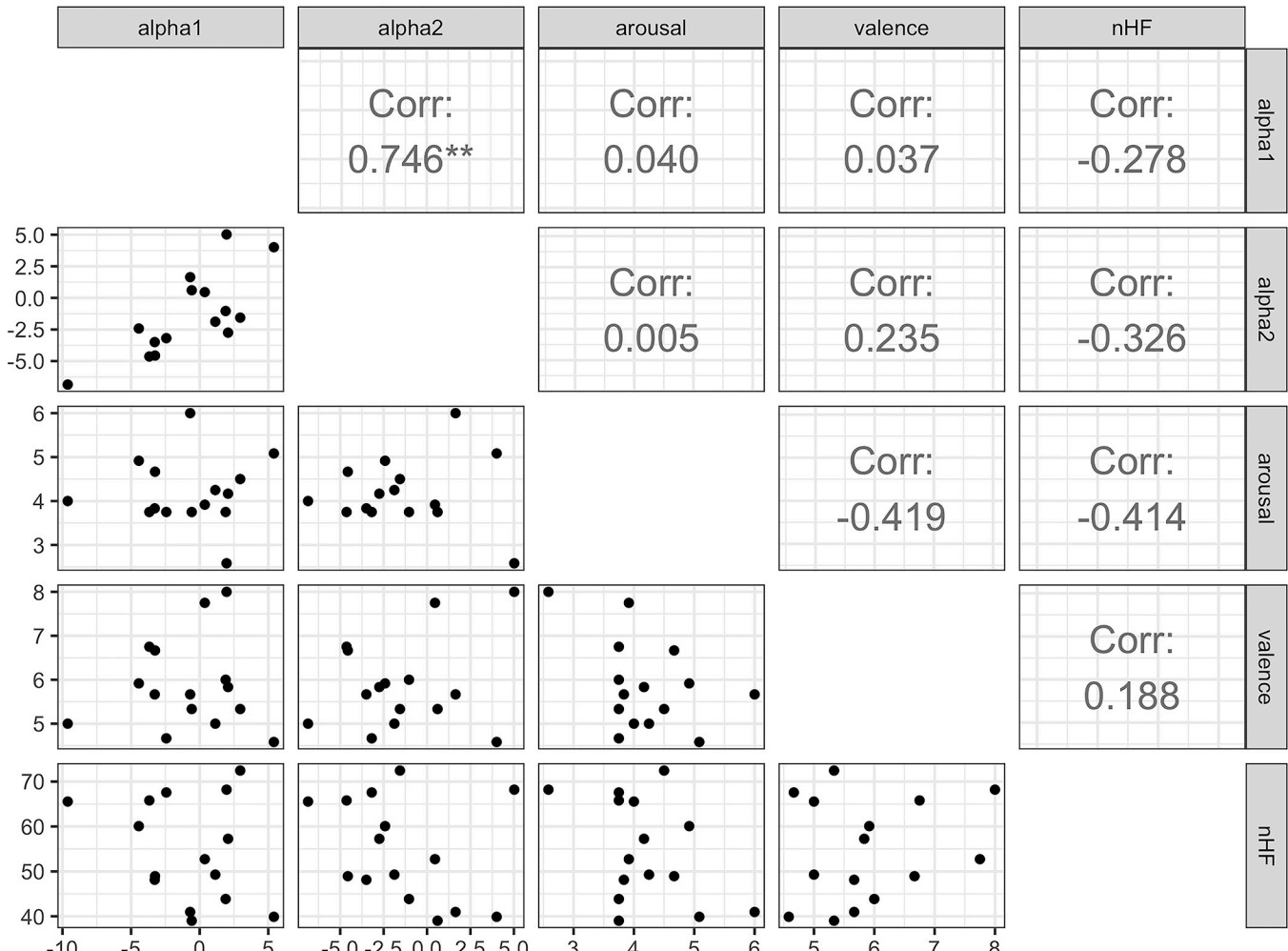

**Fig 4. Pearson correlations between subjective measures and EEG power of alpha1 and alpha2 during music exposure.** Each dot represents each participants' value during listening to the music.

by other variables. Therefore, in order to identify the common brain-heart interplay across different biological states, a graphical analysis using partial correlation analysis was performed. Fig 5 shows the results of graphical modeling for resting data. A positive partial correlation between nHF (index of parasympathetic activity) and sleepiness and a negative partial correlation with alpha1 and mental fatigue ratings were observed. For the alpha2 component, there is a negative partial correlation with mental fatigue ratings while parsing out the effects of nHF and sleepiness. Furthermore, a positive partial correlation between nHF and sleepiness ratings was observed. A positive partial correlation between alpha2 and nHF in the resting data was also observed.

Fig 6 shows the results of graphical modeling during music exposure. The results show a positive partial correlation between emotional arousal and nHF component while parsing out the effects of alpha1 power and emotional arousal ratings. For the alpha2 component, there is a positive partial correlation between emotional arousal ratings and alpha2 power. Critically, a positive partial correlation was also observed between alpha2 and nHF, as in the case of music exposure.

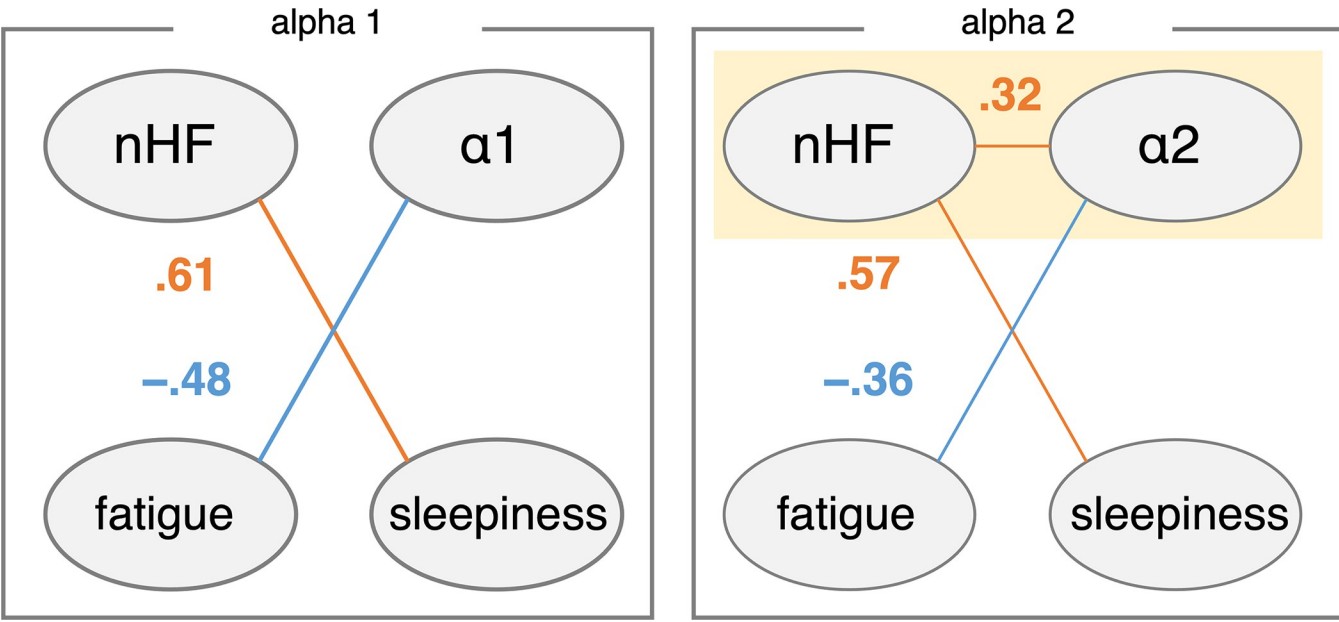

**Fig 5. Partial correlations between nodes during resting state.** Edges with nonsignificant partial correlations were removed. Note that the positive partial correlation between nHF and alpha2 power, highlighted by yellow shading, was also observed during music exposure (Fig 6).

## Discussion

This study examines the common brain-heart interplay across different biological states. A positive partial correlation was observed between arousal ratings and alpha2 components during music listening (Fig 6). This observation is consistent with recent studies showing that emotional stimuli increase alpha power [34, 35]. For example, Uusberg et al. [35] reported

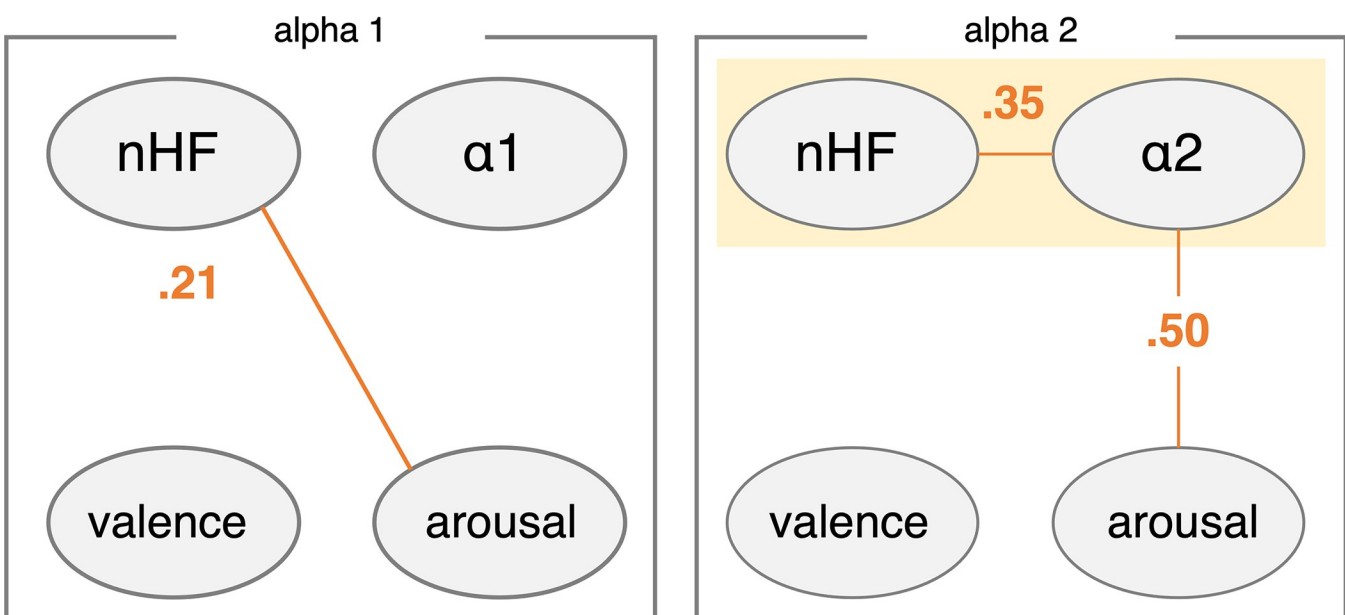

**Fig 6. Partial correlations between nodes during music exposure.** Edges with nonsignificant partial correlations were removed. The positive partial correlation between nHF and alpha2, highlighted by yellow shading, power was also observed during the resting state (Fig 5).

increased alpha power in response to emotionally aversive stimuli (high-arousal negative stimuli from the IAPS: [36]). This relationship between alpha and emotional arousal suggests an inhibitory role of alpha in the affective domain. However, according to another line of research, emotional stimulus processing is more closely associated with reduced alpha power [37, 38]. In particular, a negative correlation between arousal and alpha is often reported [39]. According to Uusberg et al. [35], the discrepancy in these findings is due to methodological differences: while participants in the study by De Cesarei and Codispoti [39] simply observed emotional stimuli, in the study by Uusberg et al. [35] as well as in the present study, they made evaluations about emotional content. Thus, the presence of goals (i.e., evaluations) that compete with focusing on emotional items may modulate the involvement of alpha oscillations. Importantly, a positive correlation between alpha and emotional arousal was observed when the other components (i.e., HRV) were separated (Fig 6). We believe that this approach would provide further insight into the direction of modulation of alpha power in emotional processing.

In the context of the resting state between the pre- and post-task conditions, a negative partial correlation was observed between the alpha2 component and fatigue ratings (Fig 5). This observation is consistent with findings demonstrating a decrease in alpha power following tasks that induce mental fatigue [19, 40, 41]. Mental fatigue is characterized as a psychological state of reduced alertness [42]. Although alpha1 has been reported to be more sensitive to mental fatigue than alpha2 [43, 44], the results of this study are generally consistent with those of studies demonstrating modulation of EEG activity by mental fatigue.

Crucially, a positive partial correlation between the alpha2 and nHF components was consistently observed in both listening and resting conditions. This relationship may represent the common brain-heart interplay across different biological states. Indeed, both higher alpha and parasympathetic activity are associated with subjective states of relaxation [45, 46]. Furthermore, increases in alpha power have been discussed as a functional inhibition of cortical activation [47, 48]. For example, increased occipital alpha power has been reported ipsilateral to the visually attended location [49, 50]. Similar changes in alpha power have been observed during working memory tasks [51, 52]. These findings suggest that alpha oscillations reflect inhibition in sensory areas that can suppress external inputs [47, 48]. The results suggest that in both biological states (i.e., listening and resting), the more relaxed participants are (as reflected by larger nHF), the more alpha power they exhibit, resulting in less external processing.

This positive correlation between parasympathetic activity and alpha power mirrors the relationship between alpha oscillations and HRV during NREM sleep, also known as quiet sleep. Ehrhart et al. [53] found that an increase in alpha power during NREM sleep was associated with a decrease in the LF/HF ratio (indicating dominant parasympathetic activity), whereas a decrease in alpha power during REM sleep was associated with a high LF/HF ratio (indicating dominant sympathetic activity). Based on these observations, Ehrhart et al. [53] suggest that alpha oscillations may reflect sleep maintenance processes. Alpha oscillations during sleep would also play a functional role in the waking state. A potential brain network supporting this interplay is the default mode network (DMN), which decreases its activity during externally oriented tasks [54, 55]. The DMN has been reported to be associated with stimulus-independent thinking or mind-wandering, which reflects decreased attention to the environment [56, 57]. In addition, the DMN is known to be influenced by cardiac and respiratory activity [58] and to show overlap with spatial patterns of alpha oscillations [59]. Deep sleep may reduce functional connectivity within the DMN [60]. Based on these empirical findings, Jerath and Crawford [61] suggest that the DMN should be an indispensable neural network for consciousness and may underlie higher-level processing. In this context, the findings of this study regarding the positive relationship between parasympathetic and alpha activity,

which is observed during both music listening and rest, would reflect this functional role of the DMN in maintaining a state of awareness or internally oriented cognition. Further research is needed to investigate the neural mechanisms underlying the brain-heart interaction.

This study has certain limitations. It has been suggested that there are multiple alpha components distributed in the human brain [62–64]. However, the relationship between these multiple alpha components and HRV remains unclear. Furthermore, this study does not address the direction of the interaction between alpha oscillations and HRV. Recently, Candia-Rivera et al. [65] proposed a bidirectional model of cortical and parasympathetic-sympathetic activities. The bidirectional interaction between alpha power and HRV warrants further investigation. Finally, since all subjects are male, the extent to which our results can be generalized to women is limited. Recent meta-analysis suggests sex differences in HRV [66], therefore it is necessary to investigate the sex difference in heart-brain interaction observed in the current study.

In conclusion, by excluding the indirect effects of subjective state on EEG and HRV activity, a direct heart-brain interplay can be observed, revealing a positive partial correlation between the alpha2 and nHF components. This direct relationship between alpha power and HRV suggests that participants who are more relaxed (as reflected by greater nHF) show less external processing (as reflected by greater alpha power) in both biological states.

## Acknowledgments

The authors would like to thank Enago (www.enago.jp) for the English language review. The manuscript was drafted using DeepL Write and Microsoft Bing AI Chat. The AI-generated text was read, revised, and proofed by the authors.

## Author Contributions

**Conceptualization:** Tomoya Kawashima, Honoka Shiratori, Kaoru Amano.

**Data curation:** Honoka Shiratori.

**Formal analysis:** Tomoya Kawashima, Honoka Shiratori.

**Funding acquisition:** Tomoya Kawashima.

**Investigation:** Tomoya Kawashima, Honoka Shiratori.

**Methodology:** Tomoya Kawashima, Honoka Shiratori, Kaoru Amano.

**Project administration:** Kaoru Amano.

**Supervision:** Kaoru Amano.

**Validation:** Tomoya Kawashima, Honoka Shiratori, Kaoru Amano.

**Visualization:** Tomoya Kawashima, Honoka Shiratori.

**Writing – original draft:** Tomoya Kawashima.

**Writing – review & editing:** Tomoya Kawashima, Honoka Shiratori, Kaoru Amano.

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
