## [Decision Letter · Decision Letter 0]

11 Dec 2023

PONE-D-23-34355The Relationship Between Alpha Power and Heart Rate Variability Commonly Seen in Various Mental StatesPLOS ONE

Dear Dr. Kawashima,

Thank you for submitting your manuscript to PLOS ONE. After careful consideration, we feel that it has merit but does not fully meet PLOS ONE’s publication criteria as it currently stands. Therefore, we invite you to submit a revised version of the manuscript that addresses the points raised during the review process. Reviewer#1 suggests rejection, while Reviewer#2 is for a minor revision. I agree with Reviewer#1 that there are notable criticisms regarding the introduction, methods sections, and data analysis that undermine the solidity of the interpretation of the results and make the publication of this manuscript unacceptable. However, I also believe that the authors can address all the suggested changes. Thus, I invite them to provide an extensive revision of all the mentioned critical sections pointed out by both reviewers and resubmit their revised manuscript for further consideration.

We look forward to receiving your revised manuscript.

Kind regards,

Vilfredo De Pascalis

Academic Editor

PLOS ONE

Journal Requirements:

SPS Grant-in-Aid for Early-Career Scientists (JP20K14274 and JP 23K17171)

This research was supported by a JSPS Grant-in-Aid for Early-Career Scientists (JP20K14274 and JP 23K17171) to T.K. The authors would like to thank Enago (www.enago.jp) for the English language review. The manuscript was drafted using DeepL Write and Microsoft Bing AI Chat. The AI-generated text was read, revised, and proofed by the authors.

SPS Grant-in-Aid for Early-Career Scientists (JP20K14274 and JP 23K17171)

Additional Editor Comments:

Reviewer#1 suggests rejection, while Reviewer#2 is for a minor revision. I agree with Reviewer#1 that there are notable criticisms regarding the introduction, methods sections, and data analysis that undermine the solidity of the interpretation of the results and make the publication of this manuscript unacceptable. However, I also believe that the authors can address all the suggested changes. Thus, I invite them to provide an extensive revision of all the mentioned critical sections pointed out by both reviewers and resubmit their revised manuscript for further consideration.

Reviewers' comments:

Reviewer's Responses to Questions

**Comments to the Author**

1. Is the manuscript technically sound, and do the data support the conclusions?

Reviewer #1: No

Reviewer #2: Yes

2. Has the statistical analysis been performed appropriately and rigorously? 

Reviewer #1: I Don't Know

Reviewer #2: Yes

3. Have the authors made all data underlying the findings in their manuscript fully available?

Reviewer #1: No

Reviewer #2: Yes

4. Is the manuscript presented in an intelligible fashion and written in standard English?

Reviewer #1: No

Reviewer #2: Yes

5. Review Comments to the Author

Reviewer #1: In the present study, titled “The Relationship Between Alpha Power and Heart Rate Variability Commonly Seen in Various Mental States”, authors investigate the relationship between Alpha Power and HRV by having participants undergo music listening sessions to evoke different mental states. Authors suggest that their results point towards evidence of an interplay between alpha and HRV. However, some major changes are in order before publication. There are some notable criticalities regarding the introduction and methods sections and in the data analysis that mine the solidity of the interpretation of the results.

Major comments

• The introduction needs to be expanded with additional crucial information that is now missing. I suggest adding a section introducing and detailing what HRV is and its relevance within the context of this manuscript. It is currently only briefly mentioned, and this information is not sufficient to support the rest of the manuscript. Furthermore, it is recommended to search the literature for studies that investigated the relationship between HRV and alertness, just like the EEG evidence reported in the first paragraph.

• The stimuli paragraph is a bit confusing. I suggest specifying that each of the eight pieces was created using a combination of six 30s samples. Moreover, the rationale for choosing these pieces should be further elaborated. Why specifically 8 of those were a combination of clips from the MER database? Were the remaining 4 pieces not given a valence-arousal value? If so, why not? Are these 4 pieces to be considered as control stimuli? Is there another study in the literature that has been used to guide the authors’ decisions? Authors write that the pieces from Naji’s work were chosen because they were less frequently heard, but based on which metrics? What is the benefit coming from the fact that these supposedly evoke no memories? I am sure the authors carefully considered these questions before implementing the study, but this information needs to be clearly spelled out for readers as well.

• Page 6, music exposure paragraph: I believe there could be a more robust way of setting up the analysis, which could provide more interesting insights. If each track is assigned a valence-arousal value so that it falls in one of the four quadrants (but I am not sure of this, as the stimuli paragraph needs more details as mentioned above), then it would be possible to have four sets of tracks each composed of three tracks. This way, authors could run an ANOVA by considering the mean of each set instead of the twelve tracks individually. Moreover, this would make sense since tracks in each subset are expected to not be different among them. Finally, if the ANOVAs were to highlight significant differences, those should be further investigated with the use of post-hoc analyses.

• Page 6, line 32: authors state “as can be seen from the individual data points shown in the graphs, there appear to be individual variation”. I would argue that this is not a very sound methodology to discern eventual differences between conditions. I suggest implementing a more considerate approach when hypothesizing differences in the data.

• The number of figures is overwhelming relative to the length of the manuscript. Some of these can be removed altogether, like figure 1, 3, and 4 since they report data with no significant differences. Figures 5, 6, and 7, provided the suggestion above relative to the analyses is implemented, could be merged into a single figure, as well as figure 10 and 11. This would bring the total number of figures to 5, which is more concise by delivering all the important information.

Reviewer #2: The authors examined the relationship between EEG (8 channels) and HRV (using ECG) in 15 male subjects during 3-minute sessions with eyes closed. They incorporated a music segment in their design to control for mental states, which might aid the study. Overall, the paper addresses an interesting question, and the analysis is appropriate and straightforward.

The overall results are negative, with a somewhat weak partial correlation between alpha2 and nHR. Nevertheless, reporting negative results is also important. As the authors mentioned, this correlation has shown inconsistency across the literature, and this study might contribute another piece to this puzzle.

Major issues:

The main results of the partial correlation are not clearly reported in the text.

1. I assume the numbers in Figures 10 and 11 correspond to these results, but it's challenging to understand what was controlled for and how. It's crucial to clarify this, especially since the correlations prior to this did not yield a significant result.

2. The term 'nHF' is not defined. How is it calculated? If it's a key result, its calculation method should be clearly explained.

3. The rationale for choosing only male participants should be explained.

6. PLOS authors have the option to publish the peer review history of their article (what does this mean?). If published, this will include your full peer review and any attached files.

Reviewer #1: No

Reviewer #2: No

---

## [Author Response · Author response to Decision Letter 0]

18 Jan 2024

The followings are our point-to-point responses to the reviewers’ comments.

Comments from Reviewer 1

The introduction needs to be expanded with additional crucial information that is now missing. I suggest adding a section introducing and detailing what HRV is and its relevance within the context of this manuscript. It is currently only briefly mentioned, and this information is not sufficient to support the rest of the manuscript. Furthermore, it is recommended to search the literature for studies that investigated the relationship between HRV and alertness, just like the EEG evidence reported in the first paragraph.

Thank you for your suggestion. We have followed your suggestion and added a definition of HRV. We have also added the following paragraph explaining that HRV reflects autonomic nervous system activity, citing the literature on the relationship between HRV and arousal level.

Heart Rate Variability (HRV) is the time variation between successive heartbeats, which is a beat-to-beat variation of R-R intervals. HRV analysis can assess the state of the autonomic nervous system [15]. The frequency feature in HRV is thought to be related to the dynamics of the sympathetic and parasympathetic nervous systems. While the low frequency (LF) band (0.04–0.15 Hz) reflects both sympathetic and vagal responses, the high frequency (HF) band (0.15–0.4 Hz) may be associated with parasympathetic control of the heart [15]. The relationship between HRV and emotional arousal has been studied. For example, Iwanaga et al. [16] showed that excited music increased the LF component of HRV, suggesting a decrease in parasympathetic nervous system activation. Another line of research reported that relaxing music increases parasympathetic nervous system activity, as reflected in HRV [17]. Thus, the quantification of HRV parameters can serve as an index of the autonomic nervous system’s status.

The stimuli paragraph is a bit confusing. I suggest specifying that each of the eight pieces was created using a combination of six 30s samples. Moreover, the rationale for choosing these pieces should be further elaborated. Why specifically 8 of those were a combination of clips from the MER database? Were the remaining 4 pieces not given a valence-arousal value? If so, why not? Are these 4 pieces to be considered as control stimuli? Is there another study in the literature that has been used to guide the authors’ decisions? Authors write that the pieces from Naji’s work were chosen because they were less frequently heard, but based on which metrics? What is the benefit coming from the fact that these supposedly evoke no memories? I am sure the authors carefully considered these questions before implementing the study, but this information needs to be clearly spelled out for readers as well.

Thank you for pointing this out. The explanation was inappropriate and has been corrected. The 12 experimental stimuli were selected aiming to cover all quadrants of emotional valence and arousal. First, for the eight experimental stimuli, six 30-second stimuli that were supposed to cover all quadrants were selected from the MER database to create a 3-minute video. Because the MER database contained a lot of music that was unfamiliar to the participants, such as ethnic music, we added 4 additional musical stimuli that we thought would be more familiar to the participants, based on Naji et al. The following modifications were made.

The experimental stimuli were selected with the aim of covering all quadrants of emotional valence and arousal. Twelve 3-minute music pieces (stimuli) that could evoke emotional responses were selected. For the eight music files, six 30s music clips were taken from the MER database [26] Because the MER database included a substantial amount of music that was unfamiliar to the participants, such as ethnic music, we supplemented it with four additional music stimuli that were familiar to them and likely to evoke emotion, based on Naji et al. [27]. 

Page 6, music exposure paragraph: I believe there could be a more robust way of setting up the analysis, which could provide more interesting insights. If each track is assigned a valence-arousal value so that it falls in one of the four quadrants (but I am not sure of this, as the stimuli paragraph needs more details as mentioned above), then it would be possible to have four sets of tracks each composed of three tracks. This way, authors could run an ANOVA by considering the mean of each set instead of the twelve tracks individually. Moreover, this would make sense since tracks in each subset are expected to not be different among them. Finally, if the ANOVAs were to highlight significant differences, those should be further investigated with the use of post-hoc analyses.

Thank you for pointing this out. The 12 stimuli were selected with the aim of covering all quadrants of emotional valence and arousal. As shown in the graph below, participants' ratings of arousal and valence for the 12 stimuli were not distributed in equal numbers in each quadrant separated by the means. Therefore, we did not perform the analysis you suggested, in which the stimuli were categorized by quadrant and averages were calculated.

Our goal was not to compare EEG and HRV between quadrants, but to distribute the arousal and valence ratings. Therefore, we report the results of the analysis as before. Note that although the main effect is significant, the post hoc analysis shows no significant difference between levels.

Page 6, line 32: authors state “as can be seen from the individual data points shown in the graphs, there appear to be individual variation”. I would argue that this is not a very sound methodology to discern eventual differences between conditions. I suggest implementing a more considerate approach when hypothesizing differences in the data.

Thank you for your suggestion. To objectively show the variation of the data, we calculated the coefficient of variation (CV) for each indicator. Since there are several indicators for which the coefficient of variation is greater than 1.0, we can say that there is a large standard deviation from the mean. On this basis, we have changed the text as follows to claim that there are large individual differences in the change of the indicators.

As noted above, no significant differences were found in alpha power, HRV, or subjective reports when compared at rest before and after the task, suggesting that changes in these variables common across participants did not exist. Here, we calculated the coefficient of variation (CV) to assess the relationship between the index and their variability relative to the mean (Table 1). The coefficient of variation is the standard deviation divided by the mean. As can be seen from some CVs in these indices that are larger than 1.0, there appears to be individual variation in the pre- and post-rest changes in these indicators.

The number of figures is overwhelming relative to the length of the manuscript. Some of these can be removed altogether, like figure 1, 3, and 4 since they report data with no significant differences. Figures 5, 6, and 7, provided the suggestion above relative to the analyses is implemented, could be merged into a single figure, as well as figure 10 and 11. This would bring the total number of figures to 5, which is more concise by delivering all the important information.

Thank you for pointing this out. For data with no significant differences, we have removed the figures and presented the results in the text. Specifically, Figure 1 (subjective ratings of fatigue and sleepiness) and Figure 4 (HRV before and after the task), as well as Figure 5 (subjective ratings for each music group), Figure 6 (EEG for each music group), and Figure 7 (HRV for each music group) were removed and the mean and standard deviation information was added to the text.

 

Comments from Reviewer 2

1. I assume the numbers in Figures 10 and 11 correspond to these results, but it's challenging to understand what was controlled for and how. It's crucial to clarify this, especially since the correlations prior to this did not yield a significant result.

Thank you for pointing out that Figures 10 and 11 (Figures 5 and 6 in the revised manuscript) show the results of calculating partial correlations. In this partial correlation analysis, we computed partial correlations that control for the influence of other indicators when analyzing correlations between two indicators. The correlation analysis in Figures 8 and 9 (Figures 3 and 4 in the revised manuscript) is not a partial correlation, and the results differ from the partial correlation analysis because the influence of other indicators is not controlled.

2. The term 'nHF' is not defined. How is it calculated? If it's a key result, its calculation method should be clearly explained.

Thank you for pointing this out. The nHF calculation method was described in the Methods section, but it was written as "normalized LF" and did not include the abbreviation (i.e., nLF), which was confusing. We have corrected this as follows.

We calculated normalized LF and HF power, defined as the relative power of each component in proportion to total power minus VLF [32]: normalized LF (nLF) as the LF power in normalized units LF/(total power − VLF) × 100 and normalized HF (nHF) as the HF power in normalized units HF/(total power − VLF) × 100. The nLF and nHF mainly reflect sympathetic and parasympathetic responses, respectively. Normalization can eliminate much of the substantial variability in raw HRV spectral power, both within and across subjects [33].

3. The rationale for choosing only male participants should be explained.

Thank you for pointing this out. There was no reason to limit the participants in this study to males. However, since gender differences were reported, especially for HRV, we have added the following as a limitation of this study.

Finally, since all subjects are male, the extent to which our results can be generalized to women is limited. Recent meta-analysis suggests sex differences in HRV [66], therefore it is necessary to investigate the sex difference in heart-brain interaction observed in the current study.

---

## [Decision Letter · Decision Letter 1]

29 Jan 2024

PONE-D-23-34355R1The Relationship Between Alpha Power and Heart Rate Variability Commonly Seen in Various Mental StatesPLOS ONE

Dear Dr. Kawashima,

Thank you for submitting your manuscript to PLOS ONE. After careful consideration, we feel that it has merit but does not fully meet PLOS ONE’s publication criteria as it currently stands. Therefore, we invite you to submit a revised version of the manuscript that addresses the points raised during the review process. Reviewer 1 is pleased with the revised manuscript, but still has two minor issues (in the HRV paragraph and Fig. 2 captions) that need to be addressed before publication. Please address these changes and resubmit for acceptance as soon as possible. Please submit your revised manuscript by Mar 14 2024 11:59PM. If you will need more time than this to complete your revisions, please reply to this message or contact the journal office at plosone@plos.org. Please include the following items when submitting your revised manuscript:A rebuttal letter that responds to each point raised by the academic editor and reviewer(s). You should upload this letter as a separate file labeled 'Response to Reviewers'.A marked-up copy of your manuscript that highlights changes made to the original version. You should upload this as a separate file labeled 'Revised Manuscript with Track Changes'.An unmarked version of your revised paper without tracked changes. You should upload this as a separate file labeled 'Manuscript'.If applicable, we recommend that you deposit your laboratory protocols in protocols.io to enhance the reproducibility of your results. Protocols.io assigns your protocol its own identifier (DOI) so that it can be cited independently in the future. For instructions see: https://journals.plos.org/plosone/s/submission-guidelines#loc-laboratory-protocols. Additionally, PLOS ONE offers an option for publishing peer-reviewed Lab Protocol articles, which describe protocols hosted on protocols.io. Read more information on sharing protocols at https://plos.org/protocols?utm_medium=editorial-email&utm_source=authorletters&utm_campaign=protocols.

We look forward to receiving your revised manuscript.

Kind regards,

Vilfredo De Pascalis

Academic Editor

PLOS ONE

Journal Requirements:

Additional Editor Comments (if provided):

Reviewer 1 is pleased with the revised manuscript, but still has two minor issues (in the HRV paragraph and Fig. 2 captions) that need to be addressed before publication. Please address these changes and resubmit for acceptance as soon as possible.

Reviewers' comments:

Reviewer's Responses to Questions

**Comments to the Author**

1. If the authors have adequately addressed your comments raised in a previous round of review and you feel that this manuscript is now acceptable for publication, you may indicate that here to bypass the “Comments to the Author” section, enter your conflict of interest statement in the “Confidential to Editor” section, and submit your "Accept" recommendation.

Reviewer #1: (No Response)

Reviewer #2: (No Response)

2. Is the manuscript technically sound, and do the data support the conclusions?

Reviewer #1: Yes

Reviewer #2: Partly

3. Has the statistical analysis been performed appropriately and rigorously? 

Reviewer #1: Yes

Reviewer #2: I Don't Know

4. Have the authors made all data underlying the findings in their manuscript fully available?

Reviewer #1: Yes

Reviewer #2: Yes

5. Is the manuscript presented in an intelligible fashion and written in standard English?

Reviewer #1: Yes

Reviewer #2: Yes

6. Review Comments to the Author

Reviewer #1: The authors have answered all my previous comments in a satisfactory manner.

However, I still have a few comments that need to be tackled before proceeding to

publication.

- Results, HRV paragraph: Here you report that after the task nLF has (M = 60.9, SD

= 15.4, CV = 0.25), but this is a typo. The mean is 40.2 and the CV is 0.38 (taken

from the file uploaded to OSF). I suggest making a double check throughout the

manuscript to make sure all data has been reported correctly.

- Figure 2 caption: change “Rectangles” to “Triangles”.

Reviewer #2: No further comments, although the differences between the correlation and the partial correlation results still needs to be well justified.

7. PLOS authors have the option to publish the peer review history of their article (what does this mean?). If published, this will include your full peer review and any attached files.

Reviewer #1: **Yes: **Simone Battaglia

Reviewer #2: No

---

## [Author Response · Author response to Decision Letter 1]

31 Jan 2024

The followings are our point-to-point responses to the comments .

Comments from Reviewer 1

- Results, HRV paragraph: Here you report that after the task nLF has (M = 60.9, SD= 15.4, CV = 0.25), but this is a typo. The mean is 40.2 and the CV is 0.38 (taken

from the file uploaded to OSF). I suggest making a double check throughout the manuscript to make sure all data has been reported correctly.

Thank you for pointing this out. As you pointed out, the values in the text were incorrect and have been corrected. 

At the same time, the notation of the statistical tests was incorrect and has been corrected. 

We have rechecked all the results in the text to ensure that the rest is correct.

- Figure 2 caption: change “Rectangles” to “Triangles”.

Thank you for pointing this out. As you pointed out, the "rectangles" in the figure caption is incorrect and has been corrected to "triangles".

---

## [Editor Report · Decision Letter 2]

2 Feb 2024

The Relationship Between Alpha Power and Heart Rate Variability Commonly Seen in Various Mental States

PONE-D-23-34355R2

Dear Dr. Kawashima,

We’re pleased to inform you that your manuscript has been judged scientifically suitable for publication and will be formally accepted for publication once it meets all outstanding technical requirements.

Kind regards,

Vilfredo De Pascalis

Academic Editor

PLOS ONE

Additional Editor Comments (optional):

The authors have addressed the two minor issues suggested by Reviewer#1. Thus, the manuscript can be accepted for publication.

Compliments for their work.
---

## [Editor Report · Acceptance letter]

23 Feb 2024

PONE-D-23-34355R2 

PLOS ONE

Dear Dr. Kawashima, 

I'm pleased to inform you that your manuscript has been deemed suitable for publication in PLOS ONE. Congratulations! Your manuscript is now being handed over to our production team.

Kind regards, 

on behalf of

Prof. Vilfredo De Pascalis 

Academic Editor

PLOS ONE